# A ShK-like Domain from *Steinernema carpocapsae* with Bioinsecticidal Potential

**DOI:** 10.3390/toxins14110754

**Published:** 2022-11-02

**Authors:** Jorge Frias, Duarte Toubarro, Gro Elin Kjæreng Bjerga, Pål Puntervoll, João B. Vicente, Rui L. Reis, Nelson Simões

**Affiliations:** 1Biotechnology Centre of Azores (CBA), Faculty of Sciences and Technology, University of the Azores, Açores, 9500-321 Ponta Delgada, Portugal; 2NORCE Norwegian Research Centre, Nygårdstangen, 5838 Bergen, Norway; 3Instituto de Tecnologia Química e Biológica António Xavier (ITQB NOVA), Avenida da República (EAN), 2780-157 Oeiras, Portugal; 43B’s Research Group, I3Bs—Research Institute on Biomaterials, Biodegradables and Biomimetics, Headquarters of the European Institute of Excellence on Tissue Engineering and Regenerative Medicine, University of Minho, 4805-017 Guimarães, Portugal

**Keywords:** entomopathogenic nematodes, excretory/secretory products, disulfide-rich peptide, ShK, insecticidal toxins, fruit fly, toxicity assay, locomotor activity

## Abstract

Entomopathogenic nematodes are used as biological control agents against a broad range of insect pests. We ascribed the pathogenicity of these organisms to the excretory/secretory products (ESP) released by the infective nematode. Our group characterized different virulence factors produced by *Steinernema carpocapsae* that underlie its success as an insect pathogen. A novel ShK-like peptide (ScK1) from this nematode that presents high sequence similarity with the ShK peptide from a sea anemone was successfully produced recombinantly in *Escherichia coli*. The secondary structure of ScK1 appeared redox-sensitive, exhibiting a far-UV circular dichroism spectrum consistent with an alpha-helical secondary structure. Thermal denaturation of the ScK1 allowed estimating the melting temperature to 59.2 ± 0.1 °C. The results from toxicity assays using *Drosophila melanogaster* as a model show that injection of this peptide can kill insects in a dose-dependent manner with an LD_50_ of 16.9 µM per adult within 24 h. Oral administration of the fusion protein significantly reduced the locomotor activity of insects after 48 h (*p* < 0.05, Tukey’s test). These data show that this nematode expresses insecticidal peptides with potential as next-generation insecticides.

## 1. Introduction

Control of insect pests in agricultural ecosystems is one of the most pressing problems in agriculture. Public concern for the potential environmental impact of the intense use of chemical insecticides and the restriction or cancelation of certain insecticides by EU policies has stimulated the search for novel, safer insecticidal compounds and the development of alternative methods for insect pest control. Insecticidal toxins derived from predators and parasitoids, such as entomopathogenic nematodes (EPNs), are of great interest as novel bioinsecticides.

*Steinernema carpocapsae* is a parasitic insect nematode that lives in symbiosis with the bacteria *Xenorhabdus* and is used to control agricultural pests because of its ability to kill infected hosts within 2–3 days [1]. In recent decades, the scientific community has studied this pathogen to control several agricultural and fruit pests [2,3,4]. Although nematodes and bacteria work together to overcome the host immune system and promote insect death, nematodes can still live without symbiotic bacteria and kill insects using a large spectrum of active molecules [5]. The pathogenicity of this organism is attributed to excretory/secretory products (ESP) released by the infective nematode. Several studies have identified the ESPs produced by this nematode. Among these, different molecular effectors have been identified, including proteases, such as serpins [6], phenoloxidase inhibitors [7], immunomodulators [8], elastase-like serine proteases [9], astacin metalloproteases [10], apoptotic serine proteases [11], and basal lamina-degrading serine proteases [9]. These findings provide relevant insights into the molecular effectors underlying the success of nematodes as biological control agents. These studies have identified individual proteins with functions that are mainly related to tissue degradation and host immune suppression.

Lu et al. (2017) performed a secretome study of EPNs where they analyzed by mass spectrometry the total ESPs produced by induced infective juveniles (IJs). In this study, 472 ESPs, including proteases and protease inhibitors, were the most abundant, as were other proteins with toxin-related characteristics, such as ShK domain-containing proteins, fatty acids, and retinol-binding proteins. This study also showed that activated nematodes secrete lethal ESPs because the mixture is toxic to different insects, such as flies (*Drosophila melanogaster*), silkworm larvae (*Bombyx mori*), and waxworm larvae (*Galleria mellonella*) [12].

In addition, these studies contributed to the identification of several ESPs that were shown to be released by IJs upon infection and contributed to the knowledge of the transcriptomic differences that occur during infective juvenile activation. However, no previous toxicity studies have used this information to investigate the specific proteins that have potential use as natural insecticidal molecules. Only putative toxic domains have been found in secretome and individual protein studies, such as ShK domain-containing proteins or some astacin metalloproteases, namely Sc-AST [10].

ShK-like domains found in the ESPs of *Steinernema carpocapsae* resemble cysteine-rich peptides that are active blockers against potassium channels, such as the ShK peptide from the sea anemone *Stichodactyla helianthus* and BgK from *Bunodosoma granulifera*. Despite their differences in amino acid sequence and overall peptide structure, the three disulfide bridges formed between cysteine residues and the relative orientation of functionally important amino acid residues are conserved, namely the dyad composed of Lys and Tyr residues that are essential for their affinity to potassium channels [13].

Most ShK domain-containing proteins found in *S. carpocapsae* have close homologues in vertebrate-parasitic nematodes [12], suggesting that this family of molecules has essential conserved functions related to parasitism. Further research is needed to determine their toxicity and understand the underlying biological mechanisms. Here, we report the expression, characterization, and toxic effects of an ShK-like domain (ScK1) from a *S. carpocapsae* astacin metalloprotease (Sc-AST). A previous study showed upregulation of the gene encoding this peptide during the parasitic stage of *S. carpocapsae* [10], suggesting its importance in nematode infection. Although this peptide is a close homologue of ShK-like peptides from other vertebrate-parasitic nematodes known to block Kv1.3 potassium channels [14], the exact mode of action of these molecules in insects remains unknown.

## 2. Results

### 2.1. Expression and Purification of Recombinant DsbC-ScK1

The sequence encoding the toxin was fused downstream of the disulfide bond isomerase C (DsbC) next to the HRV-3C protease cleavage site at the C-terminus to support the formation of disulfide bonds in ScK1, and to improve the purification of the fusion protein, a 10xHis-tag was added to the N-terminus (Figure 1a, Appendix A). The DsbC-ScK1 fusion protein was expressed in *Escherichia* (*E.*) *coli* MC1061, purified from the clarified crude extract by nickel affinity chromatography and monitored by SDS-PAGE (Figure 1b). The recombinant fusion protein was not clearly visible in the clarified lysate of induced cells. However, a dominant protein double band appeared in the first elution fraction after buffer exchange with imidazole, indicating a moderate expression of the fusion protein (soluble levels below 20 mg/L). Based on SDS-PAGE analysis (Figure 1b, lane 3 and 5) two protein bands were visualized: one with an estimated molecular weight of 31 kDa that corresponds to the DsbC-ScK1 fusion protein and another slightly smaller that corresponds to the DsbC tag alone with an estimated molecular weight of 26 kDa. The double band identified after purification corresponds to a portion of protein truncated/degraded (lower band) during the expression and purification as described by Sequeira et al., 2017 [15].

After the second affinity chromatography step, several contaminants were removed during the flow-through step, and the fusion protein was eluted with high purity (Figure 1b, lane 5).

Mass spectrometry analysis of the excised higher-molecular-weight band originated six peptides (Table 1), covering 22% of the entire protein and matching the DsbC-ScK1 protein sequence, with a significance score of 106 (scores greater than 61 are significant at *p* < 0.05). Two of the six originated peptides corresponded to the ScK1 amino acid sequence with individual ions scores > 31, indicating extensive homology. The remaining four peptides corresponded to the DsbC tag.

### 2.2. Optimization of HRV-3C Protease Cleavage and Peptide Purification

The fusion protein was not cleaved by HRV-3C protease under the commercially recommended reaction conditions, namely using a 50 mM Tris-HCl and 150 mM NaCl, pH 8.0 buffer. Since HRV-3C protease is dependent on reducing conditions to work, the addition of increasing concentrations of dithiothreitol (DTT) to the cleavage reaction was analyzed using SDS-PAGE (Appendix A). The disappearance of the 31 kDa protein band (fusion-peptide species band) after the overnight reaction indicated fusion protein cleavage and release of the ScK1 peptide.

Although the expected 4.8 kDa ShK peptide was not visible on the gel owing to the low amount loaded, the cleavage efficiency was calculated by integrating and comparing the disappearance of the fusion peptide bands on the gel (Appendix A). No cleavage of the fusion protein by HRV-3C protease was observed at 0.1 mM DTT, but an evident increase in cleavage efficiency occurred with an increase in DTT concentration from 0.5 mM to 5 mM. The optimal cleavage reaction was observed using 5 mM DTT with a calculated cleavage efficiency of 78%. Under these optimal conditions, before the cleavage reaction started, three major protein bands were observed in the SDS-PAGE gel profile (Appendix A), corresponding to the HRV-3C protease (48 kDa), fusion protein DsbC-ScK1 (31 kDa), and fusion partner DsbC (26 kDa). Only two major protein bands were observed after the overnight cleavage reaction, indicating highly efficient digestion of the fusion peptide species (Appendix A). Similar results were achieved in the presence of the reducing agent TCEP at the same concentrations (data not shown). The reducing conditions were revealed as mandatory to the cleavage of the fusion protein. The recombinant ScK1 loaded onto Superdex peptide HR10/300 GL resulted in several peaks. Electrophoretic analysis of the peaks revealed the presence of purified peptides in the fractions corresponding to the fourth peak eluted at 16.8 mL, with an estimated molecular weight of 5 kDa (Appendix A).

The intact mass measurement of the purified peptide by MALDI-TOF/TOF yielded an m/z peak of 5693, consistent with the estimated molecular weight obtained by electrophoresis (Appendix A).

### 2.3. Secondary Structure and Thermal Resistance of ScK1 Peptide

As shown in Figure 2a, the CD spectrum of ScK1 purified in the presence of TCEP presented a broad negative band at λ ~208–250 nm with minima at ~210 nm and ~225 nm. Upon the removal of TCEP through dialysis, despite the loss in intensity, the broad negative band and minima remained. Conversely, the CD spectrum of ScK1 purified in the presence of DTT is dominated by a minimum at ~200 nm and exhibited a broad positive CD band at λ ~215–250 nm (Figure 2b). The far-UV CD spectrum of ScK1 purified in the presence of TCEP was consistent with a predominantly α-helical secondary structure (hScK1). Conversely, the spectrum obtained for ScK1 purified in the presence of DTT indicates that its secondary structure consists essentially of a random coil without evidence of α-helical or β-sheet elements (rScK1).

Dye-free DSF assays allowed the determination of the thermal denaturation curves for hScK1. Monitoring the ratio of fluorescence emission at 350 nm and 330 nm allowed estimating a melting temperature (*T*_m_) of 59.2 ± 0.1 °C both from the maximum of first derivative of the thermal denaturation curve (Figure 3) and from fitting a monophasic sigmoidal curve (Inset to Figure 3).

Because dye-free DSF assays displayed no measurable curve for rScK1 (data not shown), we used dye DSF assays. At resting temperatures, the protein exhibited relatively high fluorescence intensity. However, with the temperature gradient, an increase in fluorescence emission was observed (Appendix A), and the thermal denaturation curves were best fitted with a monophasic sigmoidal function that allowed the estimation of a *T*_m_ = 51.8 ± 0.8 °C.

### 2.4. Tertiary Structure and Multiple Structural Alignments of the ScK1 Peptide

A predicted 3D model was obtained using the Phyre 2 server with estimated model confidence of 35 residues (95%) and modeled with >90% accuracy. It is noteworthy that nearly identical 3D models were obtained using AlphaFold (Figure 4).

Multiple structural alignments of the ScK1 with the Protein Data Bank (RCSB PDB) using the Dali server revealed several highly similar structures to those of other vertebrate-parasitic nematodes (Table 2).

In the case of the ScK1 Phyre2-generated model, the peptide with the highest Z-score was BmK (PDB code 2MCR) from *Brugia malayi*, with 35 residues that formed similar structures in both proteins and a low average distance between the superposed atoms (RMSD = 0.2). Two other peptides/domains were identified from nematodes with relevant Z-scores: AcK1 (PDB code 2MD0) from *Ancylostoma caninum* (Z-score = 4.5) and Acan1 domain (PDB code 6DRI) from *A. caninum* (Z-score = 4.3). Regarding the ScK1 AlphaFold-generated model, the identified peptide with the highest Z-score was Acan1 (PDB code 6DRI) from *A. caninum*, with 36 superposed residues and an RMSD of 1.3. Two other peptides/domains from nematodes with similar Z-scores were identified: NaK1 (PDB code 7L2G) from *Necator americanus* (Z-score = 5.5) and BmK (PDB code 2MCR) from *B. malayi* (Z-score = 4.9).

Furthermore, multiple structural alignments of ScK1 identified a similar domain in the *Rattus norvegicus* and a peptide in the jellyfish *Aurelia aurita*.

Although the predicted secondary structure of the ScK1 peptide resembles peptides from vertebrate parasitic nematodes (Figure 4a), where all of them are mainly composed of alpha helices, the length and exact positions of the helical elements slightly diverge. Despite the low sequence identity (~29%) between the worm ShK peptides, they shared the same disulfide framework, as demonstrated by the superimposition of their backbones (Figure 4a,b). In contrast, a higher difference was found in the secondary structure between the worm peptides and the ShK peptide from sea anemones (PDB code: 1ROO).

Notably, ScK1 differed from the other peptides in one of the amino acids that compose the dyad that participates in potassium channel blockage (Figure 4c). The amino acids superimposed on the Lys22/Tyr23 dyad in the ShK peptide are Lys23/Ile24 in ScK1, Lys23/Tyr24 in BmK1, and Lys34/Leu35 in AcK1. In the case of Acan1 and NaK1, the superimposed dyads were different for both amino acids.

The structure-based phylogenetic tree formed two main branches (Figure 5): one composed of peptides/domains from parasitic worms and the other consisting mainly of peptides from sea anemones with only a few peptides/domains from worms. The ScK1 peptide was localized in the parasitic worm branch and was closely related to BmK1, Acan1, and NaK1, as previously described (Table 2).

### 2.5. Toxic Effect of ScK1 against D. melanogaster

The toxic effect of the fusion protein and the ScK1 peptide was inspected in an in vivo injection assay using *D. melanogaster* as a model. Both the DsbC-ScK1 fusion protein and the ScK1 peptide in the helical form (hScK1) killed *Drosophila* adults by injection at a dose of 22.6 and 20.5 µM, respectively, causing approximately 60–68% mortality within 24 h (Figure 6a). In contrast, the random form of ScK1 (rScK1) did not cause significant mortality of injected flies even at a higher dose (55.3 µM), likely due to partial or full unfolding and consequently loss of function. The DsbC-ScK1 fusion protein killed *Drosophila* adults in a dose-dependent manner by injection (Figure 6b). The calculated LD_50_ was 16.9 µM per adult fly at 24 h post injection (Figure 6c).

The highest tested dose of 29.1 µM per adult resulted in 100% mortality within 24 h. At a dose of 22.6 µM per adult, the survival rate decreased to 5% within 72 h, and a lower amount of 3.2 µM triggered 50% mortality during the same time window. The injection of DsbC tag alone did not show toxicity towards *Drosophila* within 72 h post injection.

The fact that the fusion protein was toxic by injection prompted us to study the effects in *Drosophila* via ingestion. Flies fed with DsbC-ScK1 fusion protein unpurified and purified in liquid food at doses of 65 and 0.5 mg/mL, respectively, showed a significant reduction in their locomotor activity over time (Figure 7).

Although no mortality was observed during the early days of the assay with this feeding regime, adult flies appeared stunned or paralyzed for long periods. This observation was noticed after two days of tracking using the ethoscope platform, where flies treated with the DsbC-ScK1 fusion protein showed a significant reduction in locomotor activity compared to the untreated control group (Figure 7a). Locomotor activity in the treated group decreased from 40% at the beginning of the assay to less than 20% of the time moving at the end of the assay, whereas in the control group (treated with unpurified and purified forms of DsbC tag alone), the insects remained active between 40 and 60% until the end of the assay.

The same pattern was observed when the locomotor activity traces were compared between the treated and control groups (Figure 7b). The group treated with a crude extract containing the DsbC-ScK1 fusion protein (CE: DsbC-ScK1) showed decreased activity from 40% to less than 10% of the time moving by the third day, whereas the insects subjected to crude extract containing the DsbC tag (CE: DsbC tag) remained at approximately 40% of the time moving until the end of the assay. The DsbC-ScK1 either in crude extract or purified caused the same effect on treated insects. The insects treated with the purified DsbC-ScK1 fusion protein showed decreased activity over time, from 50% to less than 20% of the time moving, while the insects subjected to purified DsbC tag alone had an overall activity between 40% and 50% of the time moving until the end of the assay.

The locomotor activity of the treated flies was analyzed at different time intervals (days) to determine when the toxic effects of the ScK1 domain were significant (Figure 8a).

No significant differences were observed between the treated and control groups on the first and second days. During the third day and last days of the assay (fourth and fifth days), significant differences were found between the treated and control groups (Tukey’s test, *p* < 0.05). No significant differences were found when the mean locomotor activity was considered during the entire assay despite an evident reduction in the fraction of time spent moving.

A significant reduction in locomotor activity was observed in the last days of the assay for almost all treated individuals compared with control flies (*p* < 0.05, Tukey’s test). Boxplots supported this finding, with a wider distribution and lower median than those in the control group (Figure 8b). Several flies were completely paralyzed and died during this period.

## 3. Discussion

This is the first study to reveal the insecticidal activity of the ShK-like domain in *S. carpocapsae*. Previous transcriptomic and secretomic studies on this nematode have revealed the presence of putative toxin-like domains, such as ShK-like domains, within metalloproteases [10,12], saposin-like proteins [17], and retinol-binding proteins [12] during the post-parasitic stage. These findings may point to a role of ShK-like domains in the mode of action. Moreover, studies have reported the domains and peptides of this family in other vertebrate−parasitic nematodes and uncovered their mechanism of action as potassium channel blockers [14,18,19].

The majority of known chemical or natural insecticides acts on a single target within the insect nervous system, and the most common targets are voltage-gated sodium (NaV) channels, glutamate receptors, γ-aminobutyric acid (GABA) receptors, nicotinic acetylcholine receptors, and acetylcholinesterases [20]. Therefore, it is crucial to search for novel insecticides that act on new or underexploited targets to minimize the potential for resistance development. Potassium channels are equally important targets for insecticides because the blockage/inhibition of these channels can lead to uncontrolled hyperexcitability, followed by paralysis and death of insects [21].

Toxins from spiders, which are known to function as potassium channel blockers, have been the most studied in terms of their observed in vivo toxicity effects in insects. For example, Australian funnel-web spiders produce toxins from the κ-hexatoxin-1 family, which selectively targets insect potassium channels with high affinity. Toxins from this family are specific and present potent insecticidal activity against dipterans, moderately powerful activity against orthopterans, and weak activity against lepidopterans [20]. Although several ShK-like domains having been identified in the excreted/secreted products of *S. carpocapsae*, no studies have explored the biological activity of these molecules so far.

This study describes the toxicity of a ShK-like domain (ScK1) from *S. carpocapsae* against adult *D. melanogaster* for the first time. This domain is a cysteine-rich peptide containing six cysteine residues, which are predicted to form three disulfide bonds. Typically, venom peptides are difficult to produce by heterologous expression using different hosts because of the lack of chaperones that assist in protein refolding by forming proper disulfide bonds. The disulfide bond isomerase (DsbC) fusion partner has been reported to support disulfide bridge formation in cysteine-rich venom peptides during recombinant expression in the periplasm and cytoplasm of *Escherichia coli* [15,22]. Therefore, we selected DsbC as a fusion partner for the ScK1 domain in *E. coli* using the arabinose inducible expression system (pBAD). The yield of the soluble DsbC-ScK1 fusion protein (<20 mg protein per L of culture medium) was consistent during scale-up of the production process (data not shown). Mass spectrometry analysis of the excised band confirmed the identity and presence of the ScK1 domain within the protein, revealing that *E. coli* is an effective heterologous host for expressing this eukaryotic protein with multiple disulfide bridges. The cleavage efficiency of the fusion protein by the HRV-3C protease was substantially influenced by the presence of DTT and TCEP, which facilitated reducing conditions for cleavage. Since excessive concentrations of DTT and TCEP can reduce peptides and induce loss of folding and, consequently, loss of biological activity, we used a recommended concentration of 0.5 mM that resulted in an excellent compromise between cleavage efficiency and the amount of active peptide, as reported previously in another study [15]. Therefore, this concentration was selected for all subsequent cleavage reactions.

In addition to cleavage efficiency, we proved that the presence and nature of the reducing agents influenced the secondary structure and thermal resistance of the resulting ScK1 peptide. The far-UV CD spectra obtained for hScK1 are consistent with the PSIPRED prediction of the three helical regions. Moreover, the spectra resemble other reported CD data for similar peptides, such as the ShK toxin from the sea anemone *S. helianthus* [23] and BgK toxin from *B. granulifera* [24].

We attempted to study ScK1 thermal denaturation using dye-free DSF to monitor intrinsic tryptophan fluorescence given that ScK1 has two tryptophan residues in its sequence. Thermal denaturation curves were obtained for hScK1 but not for rScK1 after removal of the reducing agent. Since the CD spectra revealed predominantly random coils as the secondary structure of the latter protein’s preparation, its tryptophan residues are likely completely solvent-exposed and remain unaltered upon thermal denaturation. For the α-helical ScK1, a melting temperature (*T*_m_) of 59.2 ± 0.1 °C was estimated, close to the reported degradation temperature of 60 °C for the ShK peptide from sea anemone [25]. We used dye-DSF to study thermal denaturation of the rScK1 peptide. As inferred from the dye-free DSF data, the high basal fluorescence at 20 °C indicates some degree of exposure of the hydrophobic residues even at this resting temperature. The thermal denaturation curves allowed estimating a *T*_m_ of 51.8 ± 0.8 °C, which is significantly lower than that of the α-helical ScK1. As expected, the importance of the cysteine disulfide bridges to maintain the structural integrity throughout the purification is reflected not only in its secondary structure but also in its thermal denaturation properties.

The predicted tertiary structure of the ScK1 peptide resembles potassium channel-blocking peptides from other vertebrate-parasitic worms such as BmK1 (PDB code 2MCR) from *B. malayi* and AcK1 (PDB code 2MD0) from *A. caninum*. A recent study using whole-cell patch-clamp assays showed that these peptides block Kv1.3 channels [14]. Previous research on the structural analysis of peptide toxins that block Kv1.3 channels has revealed a functional dyad containing a Lys residue and an aromatic (Tyr or Phe) or hydrophobic (Leu) residue, which is important for their activity. Therefore, functional dyads are typically conserved [13]. Interestingly, ShK-like domains from nematodes tend to share the same amino acids that compose the functional dyad known to block Kv1-type channels, similar to the most studied ShK peptide from sea anemones. This functional dyad, schematically shown in Figure 4c, is identical for BmK1 and ShK, whereas the ScK1 peptide from *S. carpocapsae* differs by one amino acid. Despite the conservation of the critical Lys residue, the aromatic Tyr residue was replaced with a non-aromatic Ile residue.

The focus of this study was to determine the toxicity of ShK-like peptides from entomopathogenic nematodes against insects. However, several issues remain unaddressed, and future directions are suggested. Future work may be helpful in studying particular aspects of this family of peptides concerning their mode of action by performing electrophysiological assays (e.g., patch-clamp assays) to better understand potassium channel specificity.

In this investigation, the insecticidal capacity of the ScK1 peptide was assessed. The purified peptide presented toxicity against fruit fly, indicating that it is produced in the active form by *E. coli*. The toxic effects were demonstrated using insect injection and feeding assays. In the injection assays, the DsbC-ScK1 fusion protein and isolated ScK1 peptide caused paralysis and mortality in a dose-dependent manner. The calculated LD_50_ within 24 h was 16.9 µM (52.3 ng/fly), indicating substantial insecticidal activity. Additionally, lower doses (<9.7 µM) can trigger significant sublethal effects that may inhibit insect fitness. Despite being alive, flies injected with lower doses exhibited impaired movement over extended periods.

Similar results were observed in another study, in which ESPs were injected from activated infective juveniles of *S. carpocapsae* into adult *Drosophila* flies [12]. In comparison, ESPs injected at a dose of 20 ng per adult fly presented much higher toxicity than the fusion protein DsbC-ScK1 at the equivalent dose. The injected ESPs were consistently lethal to flies within 2 ± 6 h at a dose of 20 ng/fly, whereas DsbC-ScK1 was lethal within 24 h at a dose of 29.1 µM (90 ng/fly). The higher toxicity of ESPs may be explained by the presence of a mixture of different molecular effectors that simultaneously and synergistically act on the insects, causing paralysis and death.

The toxicity of the fusion protein produced by feeding was less severe than that produced by injection. In the first 24 h after injection, the flies started to show reduced movement and eventually died. In the feeding assays, signs of toxicity only appeared after 48 h in the treated groups despite flies being exposed to higher doses of the toxin than those from the injection assays. The locomotor activity of the treated groups started to decrease from the second day until the end of the assay, whereas flies began to die in the last days of the assay (fourth and fifth days). In conclusion, oral toxicity was low, but there was an evident reduction in locomotor activity in all treated groups.

The same difference in toxicity between injection and feeding assays was observed in another study, where it was shown that a spider venom peptide named amaurobitoxin-PI1a was toxic when injected but orally inactive against cabbage moth larvae [26]. The toxin regains oral toxicity when fused with snowdrop lectin (*Galanthus nivalis* agglutinin; GNA), a carrier that confers resistance to proteolysis by insect gut enzymes and efficiently transports the fused toxin across the gut to the hemolymph.

The in vivo toxicity effects resulted in significant changes in insect behavior, with impaired locomotion, paralysis, and eventually death. This abnormal behavior resembles the phenotype of mutated *Drosophila* with dominant-negative versions of specific potassium channels from the Kv1 family, such as the Shaker, Kv4 (Shal), and Kca1.1 (Slo1) channels.

Several studies have investigated the roles of potassium channels and their mutations in *Drosophila*. They found that the Shaker channel regulates action potentials, synaptic transmission and plasticity, information processing in photoreceptors, sleep, and flight behavior. Mutations in the Shaker channel trigger a leg-shaking phenotype [27]. Studies using the GAL4-UAS system for local expression of a dominant-negative Shal subunit in motoneurons have found that it impairs repetitive firing and locomotion in larvae and adults [28].

Finally, Slo channel mutants exhibited impaired locomotion at high temperatures, impaired flight ability, and reduced locomotor activity at normal temperatures. Interestingly, under ether anesthesia, Slo mutants show mild leg shaking, which is less pronounced than that of Shaker mutants [29].

The ScK1 peptide exhibited insecticidal activity when injected into insects but reduced activity when administered orally. The *Drosophila* digestive tract acts as a barrier against different microorganisms and their toxins. Only cry toxins from *Bacillus thuringiensis* and *Bacillus shaericus* have proven to have effective insecticidal activity via oral delivery [30].

For oral delivery of the ScK1 peptide, the first contacted tissue was the gut. As this peptide targets proteins present in nerve or muscle cells, it must traverse the gut to reach the hemocoel and other tissues to exert lethal effects. Based on this work, future studies should further explore certain aspects of the ScK1 peptide to improve its stability and delivery to insects. It is crucial to use fusion proteins that offer redox stability, resistance to external factors, and increased peptide uptake by insects. Our study suggests that new formulations of this family of peptides can be further explored in the development of next-generation insecticides.

## 4. Conclusions

Entomopathogenic nematodes such as *Steinernema carpocapsae* are relevant sources for the discovery of new insecticidal biomolecules, including disulfide-rich peptides. Unfortunately, the use of venom peptides as therapeutic or biotechnological molecules is hampered by the difficulty of producing sufficient amounts of native and functional proteins. In this study, we successfully expressed a recombinant version of the disulfide-rich ScK1 peptide fused to the DsbC isomerase, which was been reported to enhance the solubility of these peptides while promoting correct disulfide bond formation. To date, this study assessed the insecticidal capacity of ScK1. Toxicity assays showed that this peptide had a low LD_50_ and significant sublethal effects, which may inhibit insect fitness. The insecticidal activity of these molecules opens new avenues for the development of novel bioinsecticides and next-generation insecticides.

## 5. Materials and Methods

### 5.1. Materials

All chemicals and reagents were purchased from Sigma-Aldrich (Lisbon, Portugal) except for tris-(2-carboxyethyl)-phosphine (TCEP), which was purchased from Carl Roth (Karlsruhe, Germany), and molecular biology reagents, which were purchased from Thermo Fisher Scientific (Lisbon, Portugal). The sequence encoding the ScK1 domain from nucleotides 1255 to 1365 of the Sc-AST (NCBI Accession No. GU199041) was synthesized de novo with codon usage optimized for expression in *E. coli* using the ATGenium codon optimization algorithm and the gene supplied in the pUC57 vector by NZYTech Genes and Enzymes (Lisbon, Portugal). The pINITIAL cloning vector (Addgene plasmid #46858) and pBXNH3 expression vector (Addgene plasmid #47067) were gifts from Raimund Dutzler and Eric Geertsma, respectively. All primers used for cloning were synthesized by Sigma-Aldrich (Oslo, Norway). Polyclonal antibodies raised in rabbits against the ScK1 peptide were purchased from Proteogenix (Strasbourg, France), and anti-rabbit IgG peroxidase antibodies produced in goats were from Sigma-Aldrich (Lisbon, Portugal).

The *Drosophila melanogaster* wild-type strain was cultured and maintained in a laboratory incubator at 25 °C on a potato-meal-based medium [31] with a 12 h light/dark cycle, and female adults aged between 6 and 10 days were cool-anesthetized and distributed in groups of ten in separate vials before experimental treatments.

### 5.2. Expression Construct

The pBXNHD3 (herein termed p28, using the laboratory’s arbitrary ID) expression vector was generated by introducing the gene encoding the leaderless disulfide bond isomerase (LLDsbC) fusion partner between the existing His-tag and the HRV-3C protease site of thepBXNH3 (p1) vector [32] using a restriction-free cloning method. Primers used for restriction-free cloning were designed using an online tool (http://www.rf-cloning.org, accessed on 22 March 2019) [33]. The hybrid primers (Fwd:5′-CATCATCACCATCATCATCATCATAAAGATGACGCGGCAATTC-3′ and Rev:5′-CCTTGAAACAAAACTTCTAATCTAGATTTACCGCTGGTCATTTTTTGG-3′) were used in step 1 of the Rf cloning to amplify the DsbC gene (P0AEG6) using the *E. coli* genome as a template. The primary PCR conditions were as follows: 30 s at 98 °C; 35 cycles of 8 s at 98 °C, 20 s at 52 °C, and 15 s at 72 °C; and a final extension of five minutes at 72 °C.

In a second PCR, the resulting PCR product from the first reaction was used as a megaprimer for linear amplification of the p1 vector, which was used as a template. The secondary PCR conditions were as follows: 30 s at 98 °C; 30 cycles of 30 s at 98 °C, one minute at 53 °C, 2 min, and 7 s at 72 °C; and a final extension of five minutes at 72 °C. In the final step, the PCR product composed of the DsbC gene introduced to the vector backbone was treated with DpnI to remove the parental plasmid and then transformed into *E. coli* DB3.1 for plasmid replication.

### 5.3. Cloning

To clone the gene encoding ScK1 into the expression vector, a gene fragment encoding the amino acid sequence of ScK1 was first amplified by PCR from the pUC57 delivery vector using two overlapping gene-specific primers (Fwd: 5′-atatatGCTCTTCtAGTgaatgcagcgaccgcaccaacctg-3′; Rev: 5′-tatataGCTCTTCaTGCcacgcaatagccacaactcaaggcgca-3′), both containing SapI overhangs. The ScK1-encoding gene was integrated into the pINITIAL cloning vector by digesting the PCR product and the vector with SapI and ligated with T4 DNA ligase according to the FX-cloning protocol [34]. Sanger sequencing was performed using primers specific to the vector backbone (Fwd:5′-gagtaggacaaatccgc-3′; Rev:5′-tgcttcgcaacgttcaaatccgc-3′) to confirm the correct cloning of the pINITIAL construct. Finally, the ScK1-encoding gene was subcloned by mixing the p28 expression vector with pINITIAL/ScK1, digested with SapI, and ligated with T4 DNA ligase.

The p28 empty vector was generated by integrating a gene fragment encoding a GSGSGS (GS) linker in place of ccdB gene in the vector to express the DsbC tag alone in *E. coli* MC1061 cells. The GS-linker was synthesized by hybridizing two primers into one double-stranded DNA fragment containing sticky SapI overhangs to allow subsequent FX cloning procedures [35].

### 5.4. Expression of DsbC-ScK1 in E. coli and Purification of the ScK1 Peptide

Next, we transformed both p28 expression plasmids containing DsbC-ScK1 and DsbC tag alone into *E. coli* MC1061 cells grown on LB agar medium supplemented with 100 µg/mL ampicillin. The resulting colonies were inoculated with pre-cultures of 5 mL LB liquid medium supplemented with 100 µg/mL ampicillin and incubated overnight at 37 °C. A starter culture (12.5 mL) was used to inoculate 500 mL of ZYP-5052 auto-induction medium (2% tryptone, 0.5% yeast extract, 0.5% NaCl, 0.5% glycerol, 0.05% glucose, and 0.1% L-arabinose in phosphate buffer, pH 7.2) [36] supplemented with 200 µg/mL ampicillin for large-scale production of recombinant proteins. The cultures were incubated by shaking at 250 rpm at 37 °C during 3 h, and then, one batch of cells were pelleted by centrifugation at 4500 rpm for 30 min at 4 °C. The other batch of cells were left to grow at 30 °C for overnight protein expression, and cell pellets were obtained by centrifugation.

The pellets were washed with 0.8% NaCl solution and frozen at −80 °C overnight. Next, the cell pellets were thawed and lysed at 4 °C for 2 h in standard lysis buffer (50 mM Tris-HCl, pH 8.5, 50 mM NaCl, 10% glycerol, and 0.25 mg/mL lysozyme), and the cell debris was removed by centrifugation at 13,000 rpm for 10 min at 4 °C. The supernatant corresponding to the soluble fraction was collected and dialyzed against binding buffer (20 mM sodium phosphate, 0.5 M NaCl, and 20 mM imidazole, pH 7.4). Recombinant proteins were purified from the clarified supernatant by nickel affinity chromatography (HisTrap HP, GE Healthcare, Chicago, IL, USA) under the recommended conditions in an ÅKTA FPLC system (GE Healthcare). The eluted fractions were buffer-exchanged using a PD-10 desalting column (GE Healthcare). We attempted the cleavage of the fusion protein by HRV-3C protease (Thermo Scientific 88946, Waltham, MA, USA) under different reaction conditions in the presence of dithiothreitol (DTT) and TCEP reducing agents. After cleaving the recombinant protein with HRV-3C protease, the sample was subjected to a second Ni-NTA agarose purification step. The ScK1 peptide-containing flow-through was loaded onto an analytical Superdex peptide gel filtration column (HR10/300 GL, Merck 504165) equilibrated with 100 mM ammonium acetate buffer (pH 7.0). The fractions containing the pure target peptides (>95% purity) were combined and stored at −20 °C.

### 5.5. Electrophoresis and Immunodetection of Protein Expression

The supernatant and purified fractions were denatured using urea sample buffer (25 mM Tris-HCl pH 6.8, 0.8% SDS, 3.5% glycerol, 4 M urea, 2% β-mercaptoethanol, and 0.08% bromophenol blue). All samples were run in Tris-Tricine SDS-PAGE with a 16% polyacrylamide resolving gel and a 10% stacking gel on a Mini-Protean II gel system (Bio-Rad, Hercules, CA, USA), for the analysis of protein expression. Proteins were electroblotted onto polyvinylidene difluoride (PVDF) membranes using a Mini Trans-Blot Cell (Bio-Rad) to perform immunodetection of the ScK1 domain. The membranes were washed three times in distilled water and blocked in TBST (0.01 M Tris-HCl, pH 7.5, 0.1 M NaCl, and 0.05% (*v*/*v*) Tween 20) containing 3% (*w*/*v*) BSA for 1 h at room temperature and then incubated with anti-ScK1 polyclonal antibodies raised in rabbit (1:500) in blocking solution overnight at 4 °C. Membranes were washed thrice for 10 min in TBST and incubated with goat anti-rabbit IgG peroxidase conjugate (Sigma, Lisbon, Portugal) diluted 1:5000 in TBST containing 1% (*w*/*v*) BSA. 3,3,5,5-tetramethylbenzidine was used as a substrate for the horseradish peroxidase antibody conjugate, in which a blue/brown reaction was developed on the immunoblot spots on the PVDF membrane.

### 5.6. Mass Spectrometry Analysis

The protein separated in SDS-PAGE was excised and reduced, alkylated with iodoacetamide, and digested overnight with trypsin (Promega, 10 ng/μL) at 37 °C. Tryptic peptides were desalted and concentrated using a POROS C18 (Empore, 3M) and eluted directly onto a MALDI plate using 1 μL of 5 mg/mL alphacyano-4-hydroxycinnamic acid (Sigma, St. Louis, MO, USA) in 50% (*v*/*v*) acetonitrile and 5% (*v*/*v*) formic acid. Data were acquired in positive reflector MS and MS/MS modes using a 5800 MALDI-TOF/TOF (AB Sciex, Shinagawa City, Japan) mass spectrometer and TOF/TOF Series Explorer Software v.4.1.0 (Applied Biosystems, Porto, Portugal). External calibration was performed using CalMix5 (Protea). The twenty-five most intense precursor ions in the MS spectra were selected for the MS/MS analysis. The raw MS and MS/MS data were analyzed using Protein Pilot Software v. 4.5 (AB Sciex, Framingham, MA, USA) with the Mascot search engine (MOWSE algorithm). The search parameters were as follows: monoisotopic peptide mass value, maximum precursor mass tolerance (MS) of 50 ppm, and maximum fragment mass tolerance (MS/MS) of 0.3 Da. The search was performed against a custom UniProt protein sequence database with taxonomic restrictions to *Steinernema*, containing the DsbC-3C-ScK protein sequence. Carboxymethylation of cysteines was set as a fixed modification, and the oxidation of methionine and N-Pyro Glu of the N-terminal Q were set as variable modifications. Protein identification was only accepted when significant protein homology scores were obtained, and at least one peptide was fragmented with a significant individual ion score (*p* < 0.05).

For intact mass measurement of the purified peptide, the protein solution was desalted and concentrated using POROS C8 (Empore, 3M) and eluted directly onto a MALDI plate using 1 µL of 10 mg/mL sinapic acid (Sigma) in 50% (*v*/*v*) acetonitrile and 5% (*v*/*v*) formic acid (LC-MS grade, Fisher, Lisbon, Portugal). The data were acquired in linear mid-mass positive mode using a 5800 MALDI-TOF/TOF (AB Sciex) mass spectrometer and TOF/TOF Series Explorer Software v.4.1.0 (AB Sciex). Raw MS data were analyzed using Data Explorer Software v. 4.11 (AB Sciex). External calibration was performed using a Protein MALDI-MS Calibration Kit (MSCAL3, ProteoMass, Lisbon, Portugal).

### 5.7. Far-UV Circular Dichroism and Differential Scanning Fluorimetry

Far-UV circular dichroism analysis of purified ScK1 was performed using a Jasco J-815 spectropolarimeter equipped with a CDF-426S Peltier temperature controller. Purified ScK1 was diluted to 0.15 mg·mL^−1^ in 20 mM KPi buffer (pH 7.4) and in 100 mM NaPi buffer (pH 7.4), with or without 0.5 mM TCEP. Spectra were acquired in a 0.1 cm path quartz cuvette at 20 °C as follows: eight accumulations, 50 nm·min^−1^ scan rate; data pitch 0.1 nm; data integration time, 8 s; bandwidth, 2 nm; nitrogen flow, 8 L·min^−1^. Differential scanning fluorimetry (DSF) assays were performed either in a NanoTemper Prometheus NT48 (dye-free assays; dye-free DSF) or in an Applied Biosystems QuantStudio™ 7 Flex Real-Time PCR (assays containing protein dye; dye DSF). Whereas dye-free DSF monitors protein thermal denaturation by following intrinsic tryptophan fluorescence, dye DSF monitors the thermally induced exposure of buried hydrophobic residues. Dye-free DSF assays were performed by directly transferring ScK1 at 0.15 mg·mL^−1^ (or buffer as control) into 10 μL optical capillaries (NanoTemper). The capillaries were placed in the respective holder, a linear temperature gradient between 20 °C and 90 °C at 1 °C·min^−1^ was applied, and protein unfolding was monitored by recording fluorescence emission at 330 nm and 350 nm (fluorescence excitation at 275 nm). Dye DSF assays were optimized in terms of protein concentration (50–100 µg·mL^−1^) and fluorescent dye concentration (1–2×; the Protein Thermal Shift Dye from Applied Biosystems is commercially available as a 1000× working solution). The protein–dye mixtures in a 20 μL total volume were prepared directly in a 96-well plate in the following order: buffer, dye, and protein; and the microplate was sealed with an optical adhesive cover. A linear temperature gradient from 20 °C to 90 °C at 1 °C·min^−1^ was applied and protein unfolding was monitored in the ROX channel (λ_excitation_ = 580 ± 10 nm and λ_emission_ = 623 ± 14 nm). Data from triplicate curves were averaged, and the melting temperature (*T*_m_) was determined from the maximum of the first derivative of the fluorescence vs. temperature curves, which were fitted to a monophasic sigmoidal curve using GraphPad Prism (GraphPad Software Inc., San Diego, CA, USA).

### 5.8. In Silico Prediction of ScK1 Tertiary Structure

A 3D model of the ScK1 peptide structure was obtained using the Phyre2 pipeline algorithm, a web-based bioinformatics server, to predict protein structure using homology-based modelling [37]. The web-based server encompasses four stages for determining the structural model. In the first stage, the HHblits algorithm [38] was used to determine the evolutionary profile of the amino acid sequence query by looking for residue preferences at each position, searching for homologous sequences, and generating multiple sequence alignment (MSA). The PSIPRED algorithm [39] was used to predict the secondary structure using the MSA profile, and then, both were combined into a query hidden Markov model (HMM). In stage two, the HMM information was scanned against a database of HMMs of known structures (fold library) using the HHsearch alignment algorithm [40]. The highest-scoring alignments were then used to generate raw backbone models containing insertions and deletions (indels) but without side chains. In stage three, the modelling of loops was performed by fitting fragments from a known structure database with indels of the backbone, and finally, in stage four, the side chains were fitted to the backbone generated in the previous stage by using the R3 protocol [41] to generate the final 3D structural model of ScK1. Alternatively, 3D models of Sck1 were generated using the artificial intelligence system AlphaFold [42]. Molecular representations were prepared using PyMOL (The PyMOL Molecular Graphics System, Version 2.3.3 Schrödinger, LLC, Portland, OR, USA; available at http://www.pymol.org, accessed on 11 May 2020) and UCSF Chimera Version 1.16 [43].

### 5.9. Multiple Structural Alignments and Phylogenetic Analysis

The ScK1-predicted 3D structural models were used to identify the best ShK-like peptide/domain matches from other organisms by 3D superimposition using the Dali server [44], a network service for comparing tertiary protein structures with those in the Protein Data Bank (RCSB PDB). Z-score was used to sort the list of identified peptides/domains. Similarities with Z-scores < 2 were not considered.

Amino acid sequence/profile HMM of the ScK1 domain were used to search for related ShK-like peptides in sea anemones and other parasitic worms in humans, cattle, dogs, goats, sheep, and rodents. Estwise is part of the Wise2 package [45], which allows the comparison of the protein profile HMM against the expressed sequence tag (est) databases of nematodes and anemones from NCBI. The identified related protein sequences (Appendix A) with unknown tertiary structure were batch processed using the Phyre2 server to predict the protein 3D model, and only models with an estimated confidence > 90% were selected for structure-based phylogenetic analysis (Appendix A). Multiple comparisons and 3D alignments of all protein structures were performed using the PDBeFold service on the European Bioinformatics Institute website [46], and the resulting distance matrix of the Q-score (Appendix A) was used to calculate the distance-based phylogenetic tree using the neighbor-joining method in Mega X [16].

### 5.10. Toxicity Assay of ScK1 by Microinjection in D. melanogaster

To study the toxicity of the ScK1 domain, adult female *D. melanogaster* aged between 6 and 10 days were anesthetized by cooling on ice for 10 min and then injected (total volume of 100 nL) with varying doses of recombinant DsbC-ScK1 protein and ScK1 peptide in phosphate-buffered saline (pH 7.4) using a Nanoject II Auto-Nanoliter Injector (Drummond Scientific Company, Broomall, PA, USA). The flies were allowed to recover in empty plastic tubes and were then transferred to the standard *Drosophila* solid medium. Three replicate groups of ten fly adults were injected for each dose. Mortality was recorded at 0, 12, 24, 36, 48, 60, and 72 h post injection. Flies from the control group were injected with the DsbC tag alone in phosphate-buffered saline (pH 7.4) and monitored simultaneously as the treated flies. The microinjection experiments were repeated thrice.

### 5.11. Toxicity Assay of ScK1 by Feeding in D. melanogaster

Female flies aged 6–10 days were transferred to transparent plastic tubes (70 mm × 5 mm) placed in a standard ethoscopic arena (10 × 2), and each tube was sealed with a cotton plug at one end and parafilm at the other end outside the arena. Glass microcapillary tubes were inserted vertically at the end of each tube to supply the liquid food. The flies were starved for 4 h to encourage liquid feed consumption and allowed to acclimatize to their new environment during the first day before the start of the assay. At the beginning of the assay, flies were subjected to liquid feed medium (LFM) composed of 1% sucrose, 1% skim milk, and 6% blue food dye, supplemented with 500 µg/mL purified DsbC-ScK1. The other group was treated with the crude extract (65 mg/mL) of induced cells expressing the DsbC-ScK1 protein. Two control groups were used in the assay. One control group was treated with LFM supplemented with 500 µg/mL DsbC tag, and the second control group was treated with LFM supplemented with the crude extract of induced cells containing DsbC tag alone at a final concentration of 65 mg/mL. For each fly, 15 µL of each mixture was supplied daily through microcapillaries. Twenty biological replicates per treatment and control group were used for this assay. All behavioral activities of treated and control flies were recorded for five consecutive days at a controlled temperature of 25 °C using an ethoscope [47], an open platform recording device. Ethoscopic experiments were performed with five biological replicates per treatment and repeated four times, with similar results. Twenty biological replicates per treatment and control group were used for this assay.

### 5.12. Data Analysis and Statistics

Mortality data responses from the microinjection assay within 24 h were normalized, and the concentration values were log-transformed. A nonlinear regression model (log(inhibitor) vs. normalized response) obtained using GraphPad Prism 8 software was used to estimate the LD_50_ value. The injection experiments were performed in triplicate and repeated thrice. All data obtained from the ethoscope recordings were analyzed using R software with the Rethomics package (https://github.com/rethomics, accessed on 10 June 2019). All data were preprocessed using the sleep-annotation function of the sleepr package, and activity data were analyzed in windows of 10 s. For each fly, position in the tube, maximal velocity, beam crosses, movement, and sleep annotations were obtained. Sleep was defined as whether the animal was inactive for more than five minutes. Statistical analysis consisted of one-way ANOVA for significance, and Tukey’s multiple comparison test was performed on the aggregated data.

Differences were considered statistically significant at *p* < 0.05. Outliers and dead flies during the initial acclimatization period were excluded. For locomotor activity plots, any movement detected within the 10 s time window was considered for each treated and control group. Locomotor activity traces and plots were generated in R using the ggplot2 software. Survival plots and locomotor activity boxplots were constructed using the GraphPad Prism Version 8 software.

## Figures and Tables

**Figure 1 toxins-14-00754-f001:**
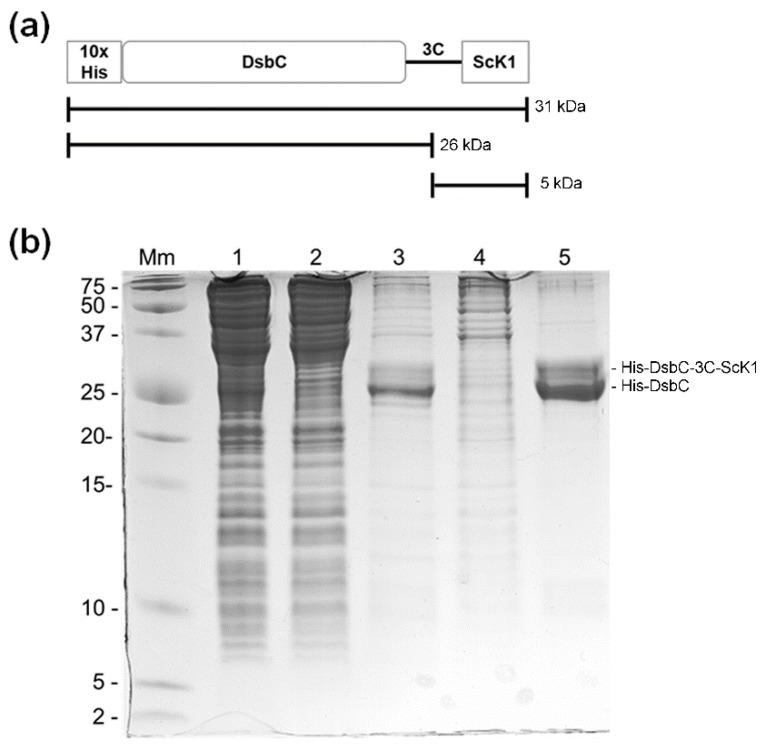
Purification of the DsbC-ScK1 fusion protein by immobilized metal affinity chromatography (IMAC). (**a**) Schematic representation of the DsbC-ScK1 fusion protein with a full-sized molecular mass and expected products after cleavage by the HRV-3C protease. (**b**) Tris-Tricine SDS-PAGE gel showing expression of DsbC-ScK1 fusion protein in *E. coli* MC1061 and the different steps of the purification protocol. Lane Mm, molecular weight protein standards (Mw in kDa); lane 1, clarified lysate after sonication and centrifugation of induced cells carrying the p28:ScK1 expression vector; lane 2, flowthrough from first Ni-NTA agarose column; lane 3, eluate from first Ni-NTA agarose column; lane 4, flowthrough from second Ni-NTA agarose column; lane 5, eluate from second Ni-NTA agarose column of DsbC-ScK1 with higher purity after buffer exchange.

**Figure 2 toxins-14-00754-f002:**
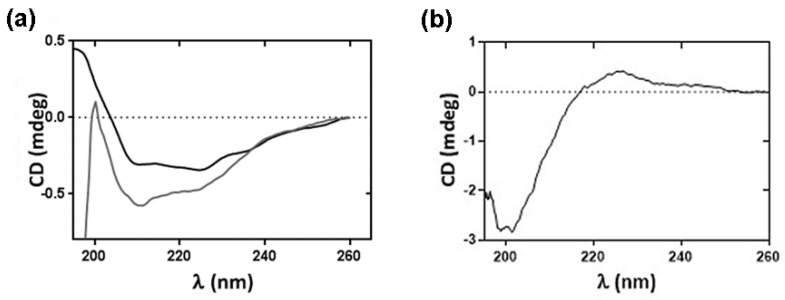
Far−UV circular dichroism (CD) analysis of purified ScK1. (**a**) CD spectrum of the ScK1 peptide purified and cleaved in the presence of 0.5 mM TCEP (grey spectrum) and after peptide dialysis for removal of TCEP (dark spectrum). Sample prepared in 100 mM NaPi buffer, pH 7.4. (**b**) CD spectrum of peptide cleaved in the presence of 0.5 mM DTT. Sample prepared in 20 mM KPi buffer, pH 7.4. Spectra result from eight accumulations at 20 °C, at a 50 nm·min^−1^ scan rate, data pitch 0.1 nm, data integration time 8 s, bandwidth 2 nm, nitrogen flow 8 L·min^−1^, and 0.1 cm light path cuvette.

**Figure 3 toxins-14-00754-f003:**
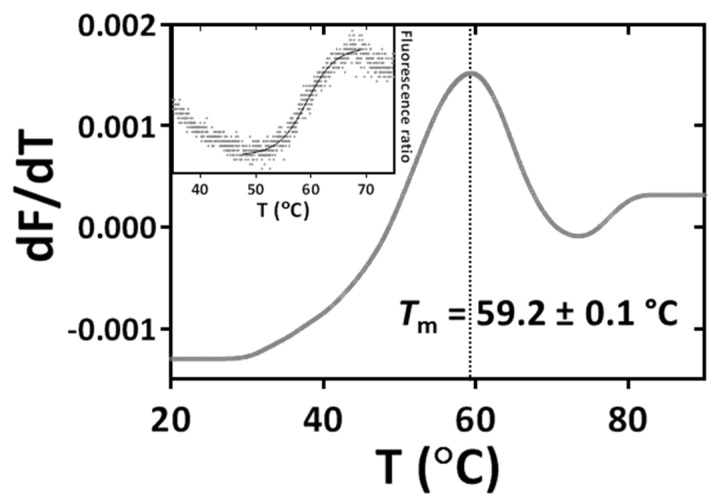
Thermal stability of the hScK1peptide. Thermal denaturation profile of hSck1, obtained by dye−free differential scanning fluorimetry (Dye−free_DSF) monitoring intrinsic tryptophan fluorescence as a function of temperature. Dye−free_DSF assays were performed in a NanoTemper Prometheus NT48. Dye−free_DSF assays were performed by directly transferring hScK1 at 0.15 mg·mL^−1^ (or buffer as control) into 10 μL optical capillaries. A linear temperature gradient between 20 °C and 90 °C at 1 °C·min^−1^ was applied, and protein unfolding was monitored by recording the fluorescence emission at 330 nm and 350 nm (fluorescence excitation at 275 nm). Data (averaged from triplicates) are represented as the first derivative of the fluorescence emission ratio (350 nm/330 nm) as a function of temperature. The melting temperature *T*_m_ = 59.2 ± 0.1 °C was determined from the maximum of the first derivative (dotted line). Inset: fluorescence emission ratio (350 nm/330 nm) as a function of temperature (black dots). Data from triplicate curves were averaged and best fitted with sigmoidal curve (variable slope) allowing to estimate *T*_m_ = 59.2 ± 0.1 °C.

**Figure 4 toxins-14-00754-f004:**
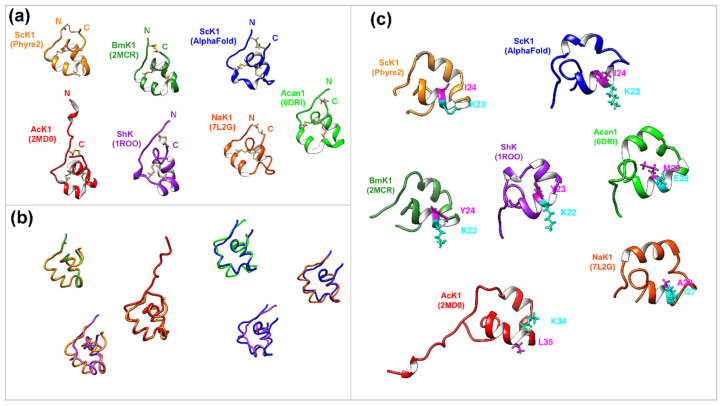
The tertiary structure of ScK1 peptide and its similarity to other parasitic nematode peptides and sea anemone peptides. (**a**) Closest-to-average structures of ScK1 Phyre2-generated model: BmK1 (2MCR) and AcK1 (2MD0); and of ScK1 AlphaFold-generated model: Acan1 (6DRI) and NaK1 (7L2G) were displayed in ribbon form to depict the secondary structure and compared with ShK (1ROO) from sea anemone. Cysteines in beige and disulfide bonds are represented by a dark line. (**b**) The superimposition of ScK1 Phyre2 and AlphaFold models (orange and blue) backbones with the closest-to-average structures BmK1 (forest green), AcK1 (red), Acan1 (green), and NaK1 (orange red) and with the ShK (magenta) from sea anemone. (**c**) Structural alignment of all peptides regarding the ShK structure using structure-based alignment in chimera. Each peptide’s identified amino acid chains correspond to the Lys22/Tyr23 dyad, which is essential for Kv1 blockage in ShK.

**Figure 5 toxins-14-00754-f005:**
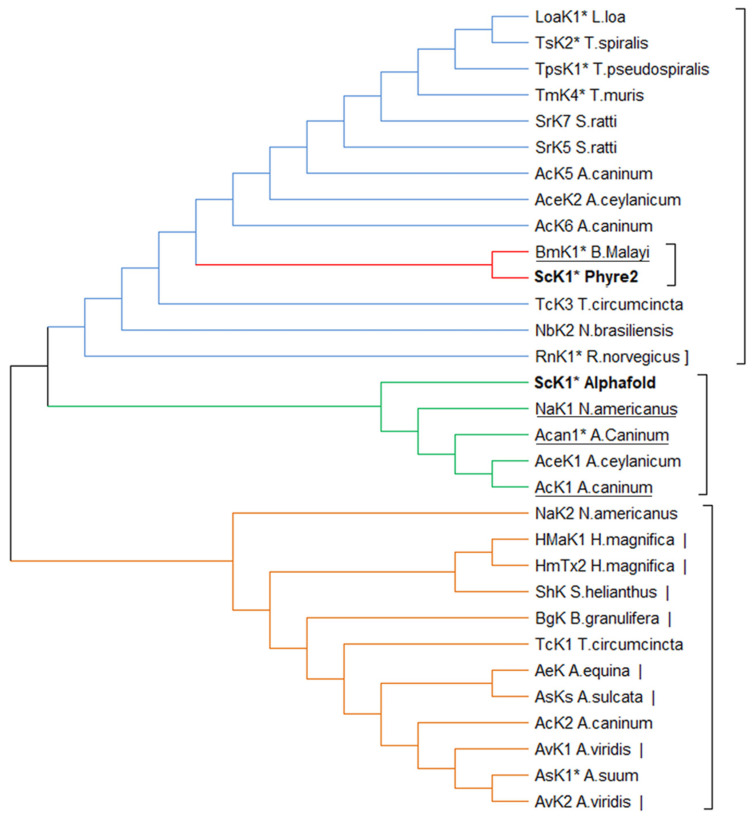
Structure-based phylogenetic tree of ShK-like peptides and protein domains from nematodes and their relationship to sea anemone toxins. Distance matrix of Q-score obtained by PDBefold was used to calculate distance-based phylogeny tree. The phylogeny was inferred using the neighbor-joining method. The optimal tree with the sum of branch length = 0.66267601 is shown. Evolutionary analyses were conducted in MEGA X [16]. The colors indicate the different main branches. The closest-to-average structures of both ScK1 models are underlined. Asterisks (*) indicate sequences that represent domains within larger proteins. Vertical lines indicate sea anemone peptides, and a square bracket indicates the vertebrate ShK-like domain.

**Figure 6 toxins-14-00754-f006:**
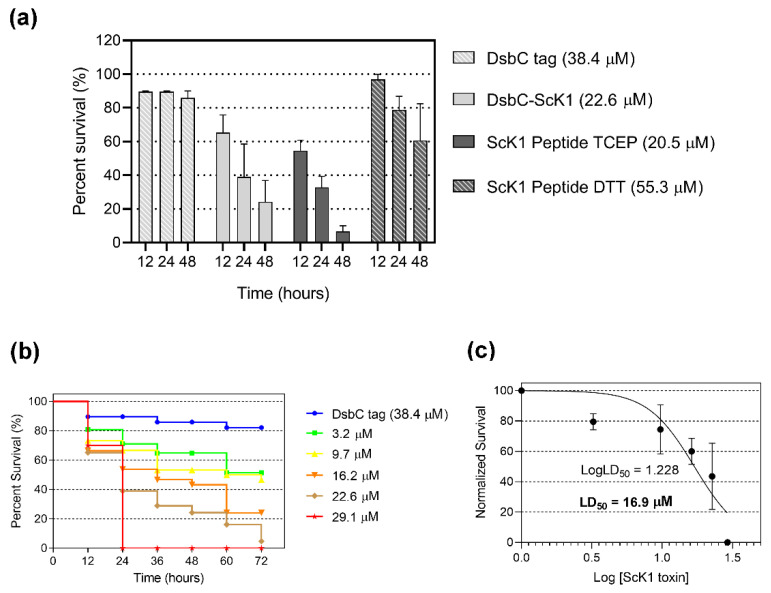
Survival rates of *Drosophila* injected with different doses of DsbC-ScK1 and ScK1 peptide. (**a**) Single-dose injection of DsbC tag (control), DsbC-ScK1 fusion protein, and ScK1 peptide isolated in the presence of TCEP and DTT, respectively, within 48 h. (**b**) Dose response of DsbC-ScK1 fusion protein injection in adults within 72 h. Control (blue); toxin doses: 3.2 µM (green), 9.7 µM (yellow), 16.2 µM (orange), 22.6 µM (brown), and 29.1 µM (red). (**c**) Normalized survival dose–response curve and LD_50_ value within 24 h of *Drosophila* injected with different doses.

**Figure 7 toxins-14-00754-f007:**
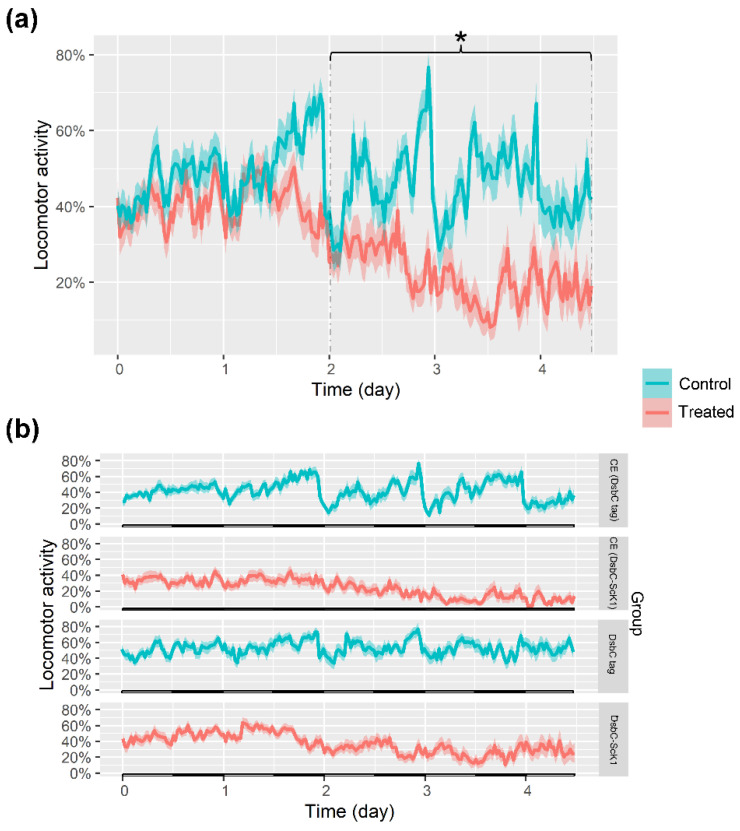
Patterns of locomotor activity. Representative traces of the movement within a 5-day time window were considered for each treated and control group. (**a**) Locomotor activity of the control group (blue) vs. treated group (red). The control group is composed of flies fed with a crude extract containing DsbC tag only and others with purified DsbC tag (*n* = 34). The treated group consists of flies fed with crude extract (CE) containing DsbC-ScK1 and others with purified DsbC-ScK1 (*n* = 36). (**b**) Locomotor activity traces of each discriminated treated and control group. The first control group was subjected to a crude extract containing DsbC tag (CE: DsbC tag) with *n* = 14) and the respective treated group with a crude extract containing DsbC-ScK1 fusion protein (CE: DsbC-ScK1) with *n*= 18). The second control group was subjected to a purified form of DsbC tag (*n* = 20) and the respective treated group with a purified form of DsbC-ScK1 (*n* = 18). The error bars are drawn as the shaded area around the trace. The asterisk (*) shows the significant differences (*p* < 0.05) between control and treated groups (Tukey’s test).

**Figure 8 toxins-14-00754-f008:**
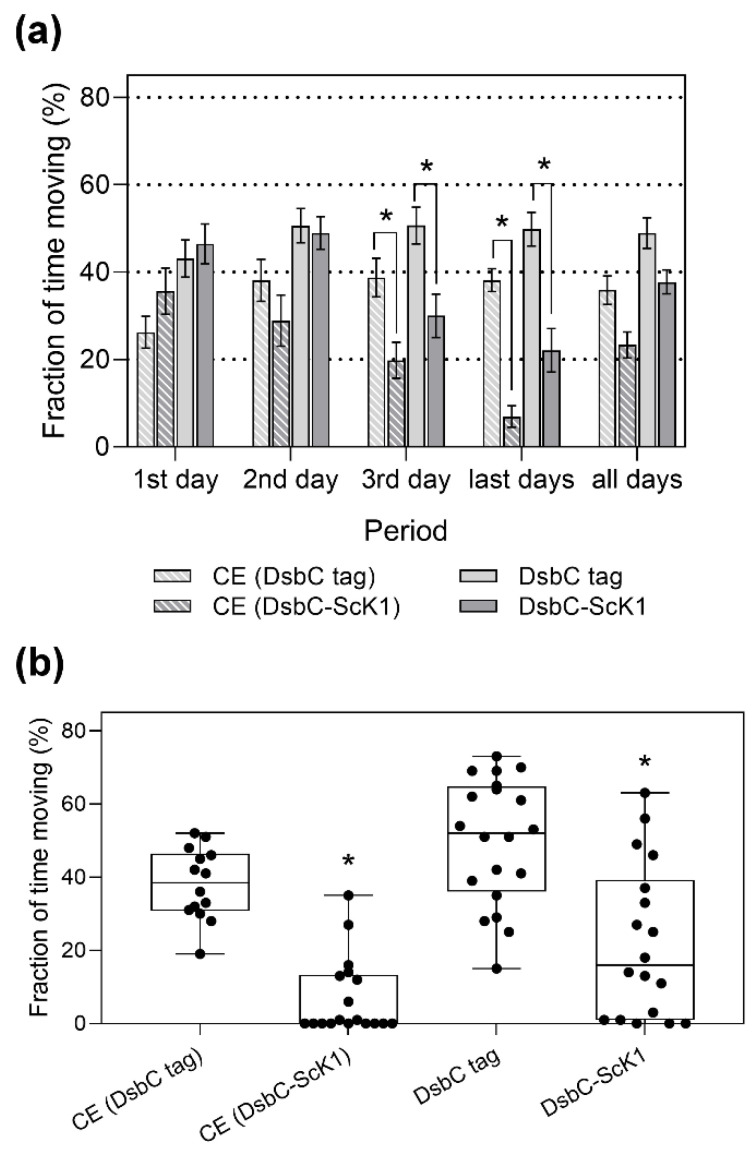
The locomotor activity of treated and control groups of *Drosophila* fed with not-purified and purified forms of the toxin and the non-toxic tag within different time intervals. (**a**) Histogram comparing locomotor activity between control groups: CE (DsbC tag) with *n* = 14 and DsbC tag (*n* = 20), with respective treated groups: CE (DsbC-ScK1) with *n* = 18 and DsbC-ScK1 (*n* = 18) within different time intervals (days). Data are presented as mean ± SEM. (**b**) Boxplots showing locomotor activity profiles of treated and control groups during the last days of the experiment. The points represent the pooled data for all the replicates of the four groups. The edges of the box denote the 25th and 75th percentiles, while the middle black line represents the median. The whiskers depict the minimal and maximal values. The asterisk (*) shows the significant differences (*p* < 0.05) between control and treated groups (Tukey’s test).

**Table 1 toxins-14-00754-t001:** Protein identification by MALDI-TOF/TOF using a custom protein database with taxonomy restriction to *Steinernema* containing DsbC-ScK1 protein sequence. The excised band originated six peptides covering 22% of the DsbC-ScK1 protein sequence with a significant score of 106. Identified peptides in bold correspond to the ScK1 domain and the other peptides to DsbC fusion protein. The protein score is −10^x^Log(P), where P is the probability that an observed match is a random event. Protein scores greater than 61 are significant (*p* < 0.05). Protein scores are derived from ions scores as a non-probabilistic basis for ranking protein hits. The asterisks (*) indicate peptides that presented an increment in mass due to the alkylation of cysteine with iodoacetamide (+57.0215 Da) or methionine oxidation (+15.9949 Da). The difference between the calculated and observed mass of 1 Da is due to (M + H)^+^ observed ions.

Total Score	Ms/Ms Peptides	Calc. Mass	Obsrv. Mass	Error ± Da	Ion Score
106	R.YLAFPR.Q	765.4	766.4	−1.028	25
	**R.SYFFEK.I**	819.3	820.4	−1.0386	31
	**R.TNLCWR.W ***	848.3	849.4	−1.0424	-
	K.EFLDEHQK.M	1044.4	1045.4	−0.9984	19
	K.LHEQMADYNALGITVR.Y	1829.9	1830.9	−1.0054	-
	K.HIIQGPMYDVSGTA PVNVTNK.M *	2256.1	2257.1	−1.0273	-

Identified peptides in **bold** correspond to the ScK1 domain.

**Table 2 toxins-14-00754-t002:** List of identified peptides/domains by multiple structural alignments or 3D superimposition loading the ScK1-generated 3D models into the Dali server. The neighbors are sorted by Z-score. Similarities with a Z-score lower than 2 are considered spurious. RMSD, root mean square deviation. The number of superposed residues (Lali), number of residues (Nres), and sequence identity (%id) are presented. Asterisks indicate sequences that represent domains within larger proteins *.

Peptide/Domain	PDB id	Species	Group	Z-Score	RMSD	Lali	Nres	%id
Phyre2-generated 3D model								
BmK *	2MCR	*Brugia malayi*	Nematode	8.1	0.2	35	36	29
RnK1 *	2K72	*Rattus norvegicus*	Rodent	4.7	1.4	35	37	31
AcK1	2MD0	*Ancylostoma caninum*	Nematode	4.5	1.5	36	51	28
Acan1 *	6DRI	*Ancylostoma caninum*	Nematode	4.3	1.4	34	40	29
Alphafold-generated 3D model								
Acan1 *	6DRI	*Ancylostoma caninum*	Nematode	5.7	1.3	36	40	31
NaK1	7L2G	*Necator americanus*	Nematode	5.5	1.0	35	39	29
Aurelin	2LG4	*Aurelia aurita*	Jellyfish	5.3	1.3	36	40	28
BmK *	2MCR	*Brugia malayi*	Nematode	4.9	1.2	34	36	32

## Data Availability

Not applicable.

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
