# Peer review of "A ShK-like Domain from Steinernema carpocapsae with Bioinsecticidal Potential"

_toxins, 2022, doi:10.3390/toxins14110754_

Round 1

Reviewer 1 Report

The manuscript "A ShK-like domain from Steinernema carpocapsae with bioinsecticidal potential" shows interesting results about a novel ShK-like peptide (ScK1) produced recombinantly in Escherichia coli that showed mortality in Drosophila melanogaster after injection and reduce locomotor activity after oral administration. I think that sounds Good and that some researchers in our field may be interested to take a look on.

The insecticidal activity of this type of molecule could open new avenues for the development of new bioinsecticides.

Author Response

We appreciate your objective assessment of the manuscript. We are encouraged by your acknowledgment of our research in terms of the results presented in the manuscript, as well as by the fact that it may inspire other scientists working in related fields to have a look at it. Thank you once again.

Reviewer 2 Report

In this MS, the authors describe that A ShK-like domain from Steinernema carpocapsae has bioin-secticidal potential. The study is interesting, howver, few key qeusitons need to clarify. 

1.Dose the ScK1 has active when expressed in E.coli. 

2. They used Drosophila melanogaster as a model to assasy toxicity assays. Whether this model really represents the toxicity analysis of Steinernema carpocapsae

Author Response

Thank you for the review of the manuscript. We considered all Reviewer's comments and we improved the manuscript regarding different sections, such as results and discussion. We also explained the use of D. melanogaster as a model to assess the toxicity of this peptide rationalizing the research design.

1.Dose the ScK1 has active when expressed in E.coli.

Thanks for pointing this out. To further clarify this issue, we removed one possibly misleading statement from the discussion “The significant amount of fusion protein produced in the active form suggested that most of the fusion protein was correctly folded.” (lines 424-426). Moreover, we added the following statement “The purified peptide presented toxicity against fruit fly indicating that it is produced in the active form by E.coli.” (lines 480-481 in the revised manuscript).

2.They used Drosophila melanogaster as a model to assasy toxicity assays.Whether this model really represents the toxicity analysis of Steinernema carpocapsae

We thank Reviewers 2 and 3 for this comment, which allowed us to further clarify the choice of Drosophila melanogaster to evaluate the peptide toxicity. We agree that there are other more severe pests such as Drosophila suzukii, Ceratitis capitata, among others. These important pests should be considered in future toxicity studies with this family of peptides.

We have chosen D. melanogaster to assess the toxicity of this peptide for several reasons:

  1. This fruit fly has been extensively used to study insect-nematode interactions (e.g. Dziedziech et al., 2020a, 2020b; Jones et al., 2022; Lu et al., 2017; Peña et al., 2015).
  2. The melanogaster is a simple insect model to study the toxic effects of several venom proteins/peptides by monitoring their behavior or mortality (e.g. Eriksson et al., 2018; Haller et al., 2017; Soares et al., 2017).
  3. Finally, and perhaps most crucially, their genetic background is well-known and will be useful for upcoming research on the insect response to this family of peptides in order to comprehend their mode of action.

References:

Dziedziech, A., Shivankar, S., Theopold, U., 2020a. Drosophila melanogaster Responses against Entomopathogenic Nematodes: Focus on Hemolymph Clots. Insects 11, 62. https://doi.org/10.3390/insects11010062

Dziedziech, A., Shivankar, S., Theopold, U., 2020b. High-Resolution Infection Kinetics of Entomopathogenic Nematodes Entering Drosophila melanogaster. Insects 11, 60. https://doi.org/10.3390/insects11010060

Eriksson, A., Anand, P., Gorson, J., Grijuc, C., Hadelia, E., Stewart, J.C., Holford, M., Claridge-Chang, A., 2018. Using Drosophila behavioral assays to characterize terebrid venom-peptide bioactivity. bioRxiv 391177. https://doi.org/10.1101/391177

Haller, S., Romeis, J., Meissle, M., 2017. Effects of purified or plant-produced Cry proteins on Drosophila melanogaster (Diptera: Drosophilidae) larvae. Sci. Rep. 7, 11172. https://doi.org/10.1038/s41598-017-10801-4

Jones, K., Tafesh-Edwards, G., Kenney, E., Toubarro, D., Simões, N., Eleftherianos, I., 2022. Excreted secreted products from the parasitic nematode Steinernema carpocapsae manipulate the Drosophila melanogaster immune response. Sci. Rep. 12, 14237. https://doi.org/10.1038/s41598-022-18722-7

Lu, D., Macchietto, M., Chang, D., Barros, M.M., Baldwin, J., Mortazavi, A., Dillman, A.R., 2017. Activated entomopathogenic nematode infective juveniles release lethal venom proteins. PLOS Pathog. 13, e1006302. https://doi.org/10.1371/journal.ppat.1006302

Peña, J.M., Carrillo, M.A., Hallem, E.A., 2015. Variation in the Susceptibility of Drosophila to Different Entomopathogenic Nematodes. Infect. Immun. 83, 1130–1138. https://doi.org/10.1128/IAI.02740-14

Soares, J.J., Gonçalves, M.B., Gayer, M.C., Bianchini, M.C., Caurio, A.C., Soares, S.J., Puntel, R.L., Roehrs, R., Denardin, E.L.G., 2017. Continuous liquid feeding: New method to study pesticides toxicity in Drosophila melanogaster. Anal. Biochem. 537, 60–62. https://doi.org/10.1016/j.ab.2017.08.016

Reviewer 3 Report

This study reports on the characterisation of an insecticidal peptides (ShK-like peptide) from the nematode Steinernema carpocapsae.

The toxicity of the peptide and the peptide fused to disulfide bond isomerase C (DsbC) was evaluated in Drosophila melanogaster showing that both were insecticidal when injected, but only the fusion protein had activity (reduction in locomotion) when delivered orally.

The manuscript is well written, data is robust, and conclusions are supported by the evidence presented.

Only a couple of minor points

There are minor grammatical errors. Some species names are not italicised.

Why was Drosophila melanogaster chosen as the target insect. Drosophila suzukii  is a major pest, would not it have been more fitting to test on this species?

Mode of action studies and stability of the peptide and fusion protein to gut enzymes would have provided important and relevant information towards the potential development of this compound as a biopesticide. However, you do suggest this as future work, which should be followed up. This, however, doesn’t distract from the wealth of information provided on the characterisation of this compound.

Author Response

Thank you for your review of the manuscript. Your valuable comments and criticisms have allowed us to improve our manuscript. We are very encouraged by your recognition of our research in terms of the presented data and conclusions in the manuscript.

There are minor grammatical errors. Some species names are not italicised.

We checked again the manuscript for spelling, grammar, punctuation, and vocabulary. All species names have been corrected for italics.

Why was Drosophila melanogaster chosen as the target insect. Drosophila suzukii is a major pest, would not it have been more fitting to test on this species?

We thank Reviewers 2 and 3 for this comment, which allowed us to further clarify the choice of Drosophila melanogaster to evaluate the peptide toxicity. We agree that there are other more severe pests such as Drosophila suzukii, Ceratitis capitata, among others. These important pests should be considered in future toxicity studies with this family of peptides.

We have chosen D. melanogaster to assess the toxicity of this peptide for several reasons:

  1. This fruit fly has been extensively used to study insect-nematode interactions (e.g. Dziedziech et al., 2020a, 2020b; Jones et al., 2022; Lu et al., 2017; Peña et al., 2015).
  2. The melanogaster is a simple insect model to study the toxic effects of several venom proteins/peptides by monitoring their behavior or mortality (e.g. Eriksson et al., 2018; Haller et al., 2017; Soares et al., 2017).
  3. Finally, and perhaps most crucially, their genetic background is well-known and will be useful for upcoming research on the insect response to this family of peptides in order to comprehend their mode of action.

References:

Dziedziech, A., Shivankar, S., Theopold, U., 2020a. Drosophila melanogaster Responses against Entomopathogenic Nematodes: Focus on Hemolymph Clots. Insects 11, 62. https://doi.org/10.3390/insects11010062

Dziedziech, A., Shivankar, S., Theopold, U., 2020b. High-Resolution Infection Kinetics of Entomopathogenic Nematodes Entering Drosophila melanogaster. Insects 11, 60. https://doi.org/10.3390/insects11010060

Eriksson, A., Anand, P., Gorson, J., Grijuc, C., Hadelia, E., Stewart, J.C., Holford, M., Claridge-Chang, A., 2018. Using Drosophila behavioral assays to characterize terebrid venom-peptide bioactivity. bioRxiv 391177. https://doi.org/10.1101/391177

Haller, S., Romeis, J., Meissle, M., 2017. Effects of purified or plant-produced Cry proteins on Drosophila melanogaster (Diptera: Drosophilidae) larvae. Sci. Rep. 7, 11172. https://doi.org/10.1038/s41598-017-10801-4

Jones, K., Tafesh-Edwards, G., Kenney, E., Toubarro, D., Simões, N., Eleftherianos, I., 2022. Excreted secreted products from the parasitic nematode Steinernema carpocapsae manipulate the Drosophila melanogaster immune response. Sci. Rep. 12, 14237. https://doi.org/10.1038/s41598-022-18722-7

Lu, D., Macchietto, M., Chang, D., Barros, M.M., Baldwin, J., Mortazavi, A., Dillman, A.R., 2017. Activated entomopathogenic nematode infective juveniles release lethal venom proteins. PLOS Pathog. 13, e1006302. https://doi.org/10.1371/journal.ppat.1006302

Peña, J.M., Carrillo, M.A., Hallem, E.A., 2015. Variation in the Susceptibility of Drosophila to Different Entomopathogenic Nematodes. Infect. Immun. 83, 1130–1138. https://doi.org/10.1128/IAI.02740-14

Soares, J.J., Gonçalves, M.B., Gayer, M.C., Bianchini, M.C., Caurio, A.C., Soares, S.J., Puntel, R.L., Roehrs, R., Denardin, E.L.G., 2017. Continuous liquid feeding: New method to study pesticides toxicity in Drosophila melanogaster. Anal. Biochem. 537, 60–62. https://doi.org/10.1016/j.ab.2017.08.016

Reviewer 4 Report

This is interesting study dedicated to analysis of the structure and insecticidal potential of novel ShK-like peptide from Steinernema carpocapsae. I have only minor issues regarding this study. 

1. Fig 2b demontrates that significant amount of DsbC-ScK1 is present by truncated product probably corresponding to the DsbC. How do you explain this effect? It will be useful if you incorporate this esplanation in the text. 

2. Fig 7 shows the patterns of locomotor activity of Drosophila. It is not clear whether the differences observed are statistically significant. Did you use any statistical apparatus to compare the samples? It would probably be useful to show error bars.

Author Response

We appreciate the suggested changes to improve the manuscript. Please find below a point-by-point response to your improvement suggestions in the results section and figures. We also attached a revised manuscript that includes the updated text and figures.

Fig 2b demontrates that significant amount of DsbC-ScK1 is present by truncated product probably corresponding to the DsbC. How do you explain this effect? It will be useful if you incorporate this esplanation in the text. 

Thank you for this comment. We added an explanation in the manuscript (lines 94 –96) that addresses this effect. We also cited a study that observed the same pattern when using DsbC as a fusion protein for the expression of disulfide-rich peptides.

Fig 7 shows the patterns of locomotor activity of Drosophila. It is not clear whether the differences observed are statistically significant. Did you use any statistical apparatus to compare the samples? It would probably be useful to show error bars.

Thanks for pointing this out. We improved figure 7 by drawing a shaded region surrounding the trace to depict the error bars. We also applied a statistical analysis (Tukey’s test) to different intervals and the difference was indicated by an asterisk. The figure legend was corrected accordingly.